# Computational modeling of the cephalic arch predicts hemodynamic profiles in patients with brachiocephalic fistula access receiving hemodialysis

Mary Hammes[1]*, Andres Moya-Rodriguez[2,3], Cameron Bernstein[4], Sandeep Nathan[5], Rakesh Navuluri[6], Anindita Basu[2]*

1 Department of Medicine, Section of Nephrology, University of Chicago, Chicago, IL, United States of America, 2 Department of Medicine, Section on Genetic Medicine, University of Chicago, Chicago, IL, United States of America, 3 Biophysical Sciences Graduate Program, University of Chicago, Chicago, IL, United States of America, 4 College, University of Chicago, Chicago, IL, United States of America, 5 Department of Medicine, Section of Cardiology, University of Chicago, Chicago, IL, United States of America, 6 Department of Radiology, University of Chicago, Chicago, IL, United States of America

* mhammes@medicine.bsd.uchicago.edu (MH); onibasu@uchicago.edu (AB)

**Data Availability Statement:** All relevant data are within the manuscript and its Supporting Information files.

## Abstract

### Background

The most common configuration for arteriovenous fistula is brachiocephalic which often develop cephalic arch stenosis leading to the need for numerous procedures to maintain access patency. The hemodynamics that contributes to the development of cephalic arch stenosis is incompletely understood given the inability to accurately determine shear stress in the cephalic arch. In the current investigation our aim was to determine pressure, velocity and wall shear stress profiles in the cephalic arch in 3D using computational modeling as tools to understand stenosis.

### Methods

Five subjects with brachiocephalic fistula access had protocol labs, Doppler, venogram and intravascular ultrasound imaging performed at 3 and 12 months. 3D reconstructions of the cephalic arch were generated by combining intravascular ultrasounds and venograms. Standard finite element analysis software was used to simulate time dependent blood flow in the cephalic arch with velocity, pressure and wall shear stress profiles generated.

### Results

Our models generated from imaging and flow measurements at 3 and 12 months offer snapshots of the patient's cephalic arch at a precise time point, although the remodeling of the vessel downstream of an arteriovenous fistula in patients undergoing regular dialysis is a dynamic process that persists over long periods of time (~ 5 years). The velocity and pressure increase at the cephalic bend cause abnormal hemodynamics most prominent along

**Funding:** This publication was made possible by the National Institute of Diabetes and Digestive Diseases (NIDDK) and the National Institutes of Health (NIH) under award number RO1DK090769. A. M-R was supported by the NSF GRFP fellowship. The work was partly funded by the Ginny and Simon Aronson Research Award, University of Chicago Institute of Translational Medicine Pilot Award, and A.B.'s research development funds.

**Competing interests:** The authors have read the journal's policy and have the following competing interests: Anindita Basu is a paid consultant for Novartis Institutes for BioMedical Research. There are no patents, products in development or marketed products associated with this research to declare. This does not alter our adherence to PLOS ONE policies on sharing data and materials.

the inner wall of the terminal cephalic arch. The topology of the cephalic arch is highly variable between subjects and predictive of pathologic stenosis at later time points.

## Conclusions

Low flow velocity and wall pressure along the inner wall of the bend may provide possible nidus of endothelial activation that leads to stenosis and thrombosis. In addition, 3D modelling of the arch can indicate areas of stenosis that may be missed by venograms alone. Computational modeling reconstructed from 3D radiologic imaging and Doppler flow provides important insights into the hemodynamics of blood flow in arteriovenous fistula. This technique could be used in future studies to determine optimal flow to prevent endothelial damage for patients with arteriovenous fistula access.

## Introduction

The arteriovenous fistula (AVF) is the recommended vascular access to provide hemodialysis for patients with end-stage renal disease (ESRD). In the United States two thirds of patients who have an AVF placed receive upper arm brachiocephalic fistula (BCF) [1]. The BCF is created by an anastomosis between the brachial artery and the cephalic vein in the upper arm (**Fig 1A**). The cephalic vein is lateral and superficial in the upper arm with a final bend called the cephalic arch (CA) that terminates at the axillary vein (**Fig 1B**). While the BCF provides a superb location for cannulation needed at every dialysis session, pathological complications of venous outflow stenosis and thrombosis, specifically of the CA, are all too common [2]. Maturation failure, meaning failure of the vein to dilate and thicken prior to use, occurs in 26–60% of patients with BCF, often due to early thrombosis [3–5]. The mechanism of why these complications occur is unknown, but past research efforts have pointed towards altered hemodynamics in the CA causing BCF failure and loss of the access site for cannulation.

The CA is an area of active and persistent geometric remodeling due to abnormal hemodynamic flow initiated with the surgical creation of the BCF. In particular, due to the surgically created anastomosis, the cephalic vein experiences pulsatile flow from the brachial artery which is transmitted up to the CA bend. As the vein dilates over time, the high flow velocities, pulsatile flow and the geometric bend of the CA, together, lead to abnormal hemodynamics in the region, including turbulent flow conditions and abnormal wall shear stress (WSS) that promote formation of stenosis, recurrent thrombosis, and eventual access failure [6–8].

Although WSS, pressure and pulsatility are important determinants of AVF maturation, the roles that these parameters play in thrombosis, stenosis and AVF failure arising in ESRD patients undergoing hemodialysis are poorly quantified as *in vivo* measurements of these parameters are problematic. Computational modeling can be a powerful engineering tool to model hemodynamic parameters in vascular systems, given the problems inherent in rigorous hemodynamic assessment and WSS derivation. Efforts to derive WSS by computational modeling have been done but only in a small number of cases [9, 10]. The longitudinal assessment of an AVF with repeated imaging and measurements as it matures and remodels over time (~months-years) is needed to understand the complications and longevity of the AVF [11].

In the current investigation, we modeled the three-dimensional (3D) geometries of the CA from five ESRD patients at two time-points- 3 and 12 months (mo.), using venogram and

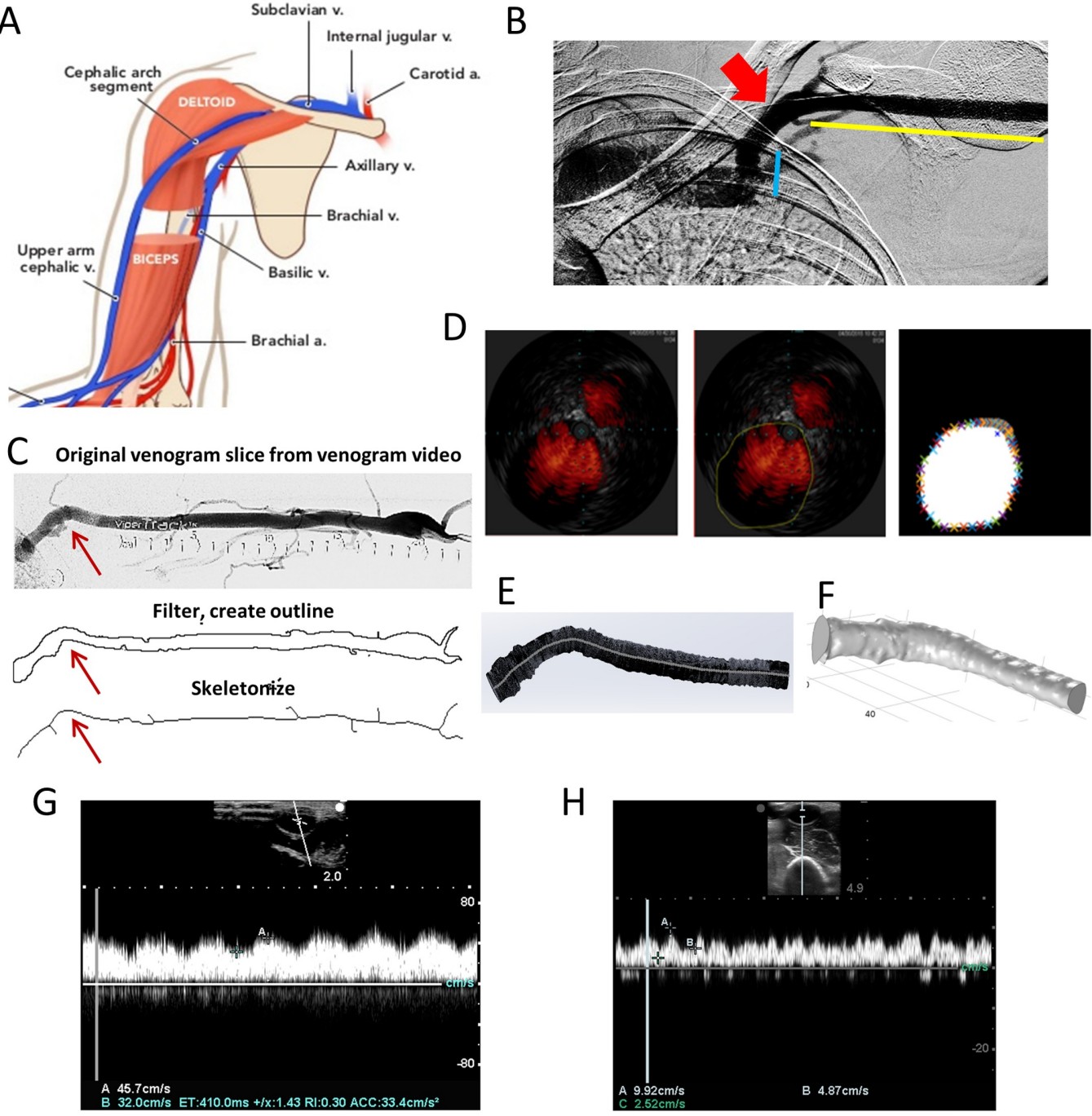

**Fig 1.** A) Schematic of the venous and arterial anatomy in the upper arm depicting the brachial artery, the cephalic vein and cephalic arch (CA). B) Venogram showing the straight cephalic vein (Pre-bend; yellow), the bend of the cephalic arch (Bend; red arrow) and straight vein (Post-bend; blue). C, D) Step-by-step processing of C) venogram, and D) IVUS images to reconstruct the 3D genometry of a patient's CA downstream of AVF placement that are used to construct internal vein contours. Red arrows in the (C) panels indicate the CA. E) Venogram and IVUS data are combined to reconstruct model of the patient's CA in 3D. F) Reconstructed model shown here is lofted and smoothed to render the 3D model used for flow simulation. G, H) Doppler trace of blood flow in a patients' CA at 3 and 12 months after BCF placement, respectively.

intravascular ultrasound (IVUS). The 3D model captures the irregularities of vein geometry that are easy to miss in two-dimension (2D), especially if these irregularities do not lie in the

plane of 2D imaging. IVUS gives a microscopic snapshot of the vein's lumen; venogram captures the CA vein path but in 2D. Combining these, we generated 3 dimensional reconstructions of the CA geometry to get a nuanced view of its remodeling over time. We then used patient-specific flow velocities and pulse rates from Doppler measurements, patient-specific whole blood viscosity and average intra-vascular pressures at the time-points of IVUS and venogram imaging to simulate patient-specific hemodynamic flow in the 3D models of the CA. We performed hemodynamic modeling in 5 patients at 3 and 12 months (mo.) after the surgical placement of the AVF. Our simulations showed regions of interest where low WSS or stenosis is likely to develop. Even as early as 3 mo. we could predict areas of abnormal hemodynamics in the CA models. These areas were likely to worsen with time and exacerbate in the same patient's CA model at the 12 mo. time-point. The overall aim of this work is to demonstrate the feasibility of computational modeling to unveil the cause of abnormal flow dynamics in the CA that are unique to individual patients. Our first hypothesis is that abnormal geometry and hemodynamics in the CA can predict areas of interest that develop into stenosis and associated pathological conditions. Our second hypothesis is that the hemodynamics modeled from actual detailed anatomic vein images are unique of each patient and differ based on the maturation of the BCF.

While attempts to computationally model vasculature for the surgical placement of an AVF [12] and track geometric and hemodynamic alterations as an AVF matures have been done [13], to our knowledge, this is the first study that models the exact 3D geometry and time-dependent flow in ESRD patients at different stages of AVF remodeling using patient- and stage-specific parameters. We note that there is significant heterogeneity between patients and between time-points which makes any conclusions made from population-averaged data difficult to interpret. Hence, we underscore the need for patient-specific models to understand the dynamic nature of CA remodeling after AVF placement, the range of possible hemodynamic conditions in the CA across patients, while predicting and therefore improving access outcomes.

## Methods and materials

### Trial design and protocol

This prospective observational trial was conducted at a single University-affiliated medical center from December 16, 2011 through April 20, 2020. This protocol [6] was approved by the Institutional Review Board from the University of Chicago (Protocol number 11–0269) on August 10, 2011. The original trial study protocol, approved prior to enrollment, is available as supporting information (see **S1 File**). The current study details the 5 subjects included as a subset for a small-scale study that had Doppler, venogram and IVUS imaging at 3 mo. and 12 mo. This trial was conducted with good clinical practice and the Declaration of Helsinki. The trial was registered at ClinicalTrials.gov (NCT 01693263) August 8, 2012. Informed consent was obtained in writing prior to any trial-related activities.

This study represents a cohort of a larger population enrolled to determine the hemodynamic consequences of the post-fistula environment. Patients were consented to enroll in the study if they had advanced renal failure or were on hemodialysis in need of a permanent access with a BCF planned. During the study 161 patients consented, but only 40 completed the study [14]. Five subjects were chosen from the 161 consented for inclusion in the study as we selected subjects with presumed normal anatomic configuration of the BCF without previous catheter access on the side of the BCF (**Fig 2**). The 5 patients included in the study were brought to the cardiac catherization lab and the following were obtained by an interventional cardiologist and radiologist sequentially on the same day: Venous blood sample for whole

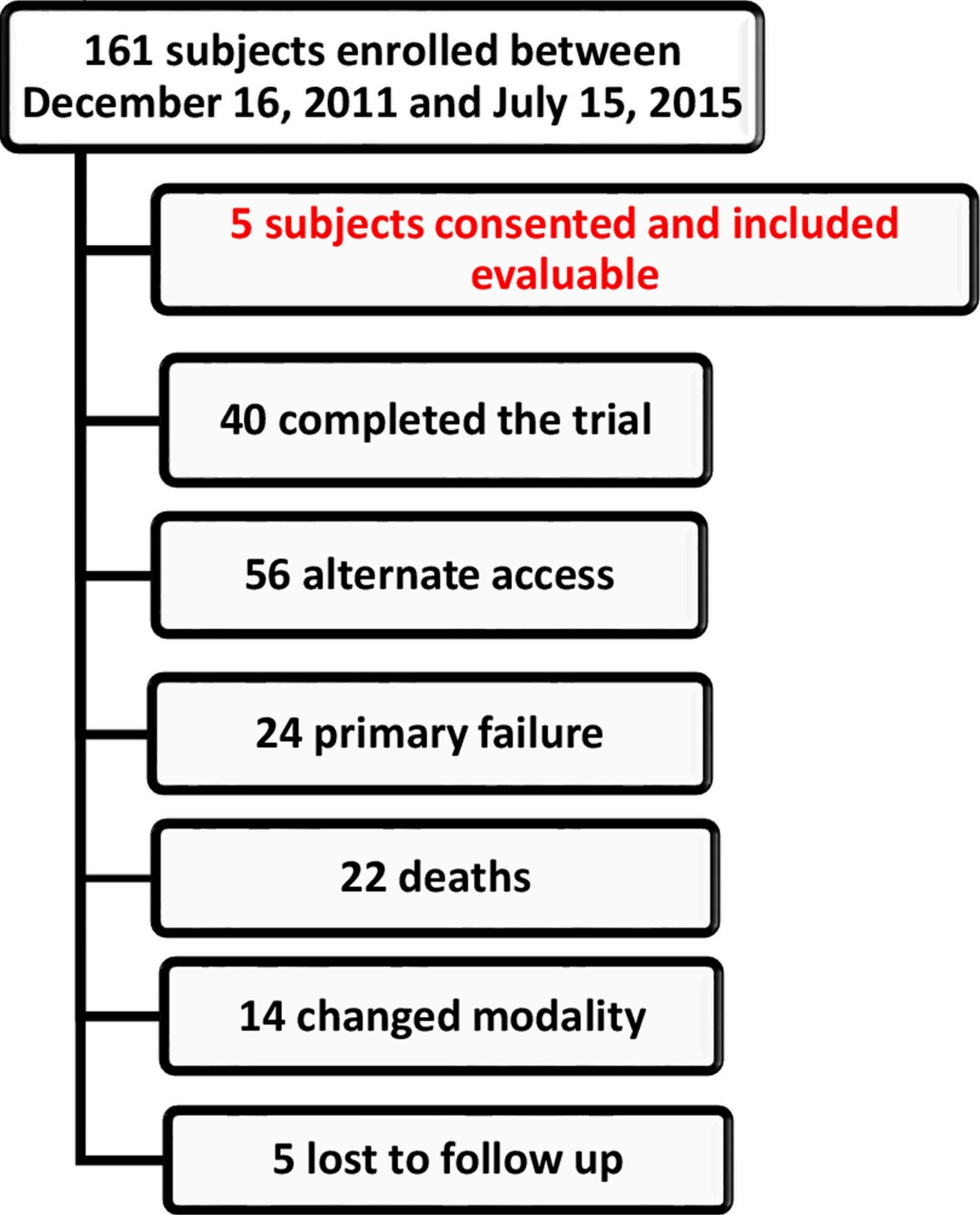

**Fig 2. Study subject inclusion and discontinuation.** Of 161 subjects enrolled, 5 subjects were suitable and agreed to participate with IVUS measurements at 3 and 12 months. 40 subjects completed the trial, 56 had alternate access placed, 24 had primary failure and never used the access for dialysis, 22 deaths occurred which were unrelated to the study, 14 subjects either received renal transplantation or transitioned to peritoneal dialysis, and 5 subjects were lost to follow-up.

blood viscosity (WBV); Doppler spectral analysis; 2D venogram imaging and IVUS of the CA. WBV was measured from venous blood samples obtained on the day of the protocol study. The blood was drawn during intravenous insertion on the day of the protocol and collected in 4.5 mL tubes containing 3.2% buffered sodium citrate. WBV was measured using a Brookfield programmable DV-II+ cone and plate viscometer. The WBV was measured at 220 s $^{-1}$ at 37˚C [15]. Three measurements of viscosity were determined at the above shear rate and the average value was used for the simulations. Spectral Doppler analysis of the CA was made at the same location in all patients. The point of measurement was marked 10 cm from the greater tubercle of the humerus as previously reported [16] which was approximately 10 cm proximal to the CA. The peak systolic velocity at a 60-degree angle of insonation was measured. The optimal Doppler angle to measure blood flow velocity is between 45 and 60 degrees [17, 18]. A 60-degree angle of insonation was used to standardize the velocity measurements as the Doppler angle of insonation has a significant effect on spectral Doppler velocity measurements [18]. Velocity measurements were taken three times, approximately 1 minute apart and then averaged. 2D venography was performed in the angiography suite by an interventional radiologist. Vascular access was obtained in the direction of the venous outflow using a micropuncture system (Cook Medical Inc., Bloomington, IN). A digital subtraction angiogram (DSA) fistulogram was then performed through the 5 French Pinnacle vascular introducer sheath (Terumo Medical Corporation, Somerset, NJ) with imaging of the venous outflow from the puncture site to the right heart. Over a 0.035-inch guide-wire, a 5 French diagnostic multipurpose angled catheter (Boston Scientific, Natick, MA) was then advanced to a straight portion of the cephalic vein approximately 10 cm downstream from the CA. A high- fidelity pressure monitor (Namic Preceptor Morse Manifold; Boston Scientific) was connected to the end of the catheter and intravascular pressure measurements were obtained. Next, a 0.014 in x 180 cm Asahi Prowater coronary guidewire (Asahi Intecc USA, Inc., Irvine, CA) was advanced through the cephalic arch under the guidance of a fluoroscopic map and IVUS was performed using a Volcano Eagle Eye Platinum 20 MHz IVUS catheter (Philips, Amsterdam NE). Images were obtained at 30 fps during automated pullback of the catheter from the axillary vein to the cephalic vein at a rate of 1 mm/sec.

## Image processing and 3D reconstruction of the cephalic arch

Image processing was performed on venogram, IVUS, and Doppler image sets with ImageJ (NIH, USA) and MATLAB (Mathworks) and 3D reconstructions of the CA for each patient were generated using Computer Aided Design (CAD) software- AutoCAD (Autodesk) and SolidWorks (Dassault Systèmes). We used a venogram recording to capture the gross anatomy of the CA, including the cephalic bend (**Fig 1C**). Briefly, all image slices were aggregated using the 'Z Project' function and 'minimum intensity' option in ImageJ to generate a 2D projection of the CA. This image was converted from gray-scale to binary (using 'Make Binary' function), cleaned to remove any features that are not associated with the CA (using 'Clear' function), inverted (using 'Invert' function) and the contour of the CA was extracted (using 'Skeletonize' function). The skeletonized image was exported in .tiff format, converted into vector graphics files (as .svg) and finally converted into a CAD file format (.dxf) that can be imported into AutoCAD and SolidWorks for 3D modeling.

IVUS images with non-directional color flow imaging (ChromaFlo), capturing 1/30 mm increments of the vein path per frame were recorded in the standard DICOM format. These pullback measurements set the lower limit of resolution along the z axis as ~1 mm. The vein contours were traced manually. The outline of the vein lumen's cross-section of every 30th image of the IVUS image stack was traced onto the image frame using ImageJ, the outline was

binarized, and the X and Y coordinates of the CA's cross-sectional outline for each slice were extracted using MATLAB, as shown in **Fig 1D**. Each slice of the IVUS at 500 x 500 pixel$^2$ or 16 x 16 mm$^2$ was thus processed and exported as individual files. The files were batch-processed using the free '*dxf2dwg*' plug-in for AutoCAD to generate corresponding .dwg files.

The area inside the traced contour in each slice was calculated in AutoCAD using the 'Extract Data' function, under the 'Annotate' tab. This step was processed in the batch-processing mode for all IVUS slices for a given time-point (3 or 12 mo.) and exported as a single text file. The mean area of cross-section and the standard deviation were calculated for all slices. The average diameter of the patient's CA was calculated from the cross-sectional area, as 2$^*$ (area/π)$^{1/2}$. The arch angle was obtained from the venogram by marking two straight sections of the brachial vein- one upstream and the other downstream to the bend, drawing tangents to the two sections, and measuring the angle inscribed between the tangents at the point where they intersect [19].

The 3D model of the CA was reconstructed by combining the global topography of the skeletonized venogram with the local topography of the contours of the vein's cross-sections from IVUS. X and Y coordinates defining each slice's cross-section were imported into AutoCAD, where the contour was properly scaled, and the geometric center of each cross-sectional slice of the vein lumen was marked. A detailed and complete 3D model of the CA was generated in SolidWorks software by importing each contour to a plane perpendicular to the path at 1 mm intervals, and aligning the center of mass of the contour with the vein path, as shown in **Fig 1E**. Using the knowledge that IVUS imaging of the CA began at the axillary vein and then moved into the CA, we utilized the rapid shift in diameters of IVUS contours to align the correct IVUS frame with where the CA model began along the skeletonized path obtained from the venogram. The digital 3D model was then lofted together across all contours, creating a smooth curved surface along the structure, shown in **Fig 1F** to replicate the continuous curvature of the vein. Designs for five subjects at 2 time-points each (3 and 12 mo.) were thus generated (**Fig 3**).

## Reynolds number

Reynolds number, $Re = \frac{\rho u L}{\eta}$ is a dimensionless parameter used to characterize flow. At low $Re$, flow is laminar but transitions to turbulent as $Re$ increases, along with the appearance of eddies and vortices. $Re$ was calculated for each subject at 3 and 12 mo. time-points, where $u$ is the peak systolic velocity measured from Doppler, $L$ is the average vein diameter measured from IVUS and $\eta$ is the WBV measured from viscosity measurements. Density, $\rho$ was set to 1060 kg/ m$^3$, the average density of whole blood [20].

## Flow simulations

We used COMSOL Multiphysics (COMSOL Inc.), version 5.5, Physics simulation software based on finite element analysis and partial differential equations [21], to simulate blood flow in the CA of patients with ESRD. In addition to the 'Multiphysics' module, the 'Microfluidics' and 'CAD Import' modules were also used. Since $Re < 2,000$ for all patients (**Table 2**), all flow were simulated using the 'Single Phase Flow' and 'Laminar Flow' physics. To test the effect of fluid flow in an idealized, simplified CA, we generated an idealized 3D vein model with a diameter of ~ 9.5 mm and a bend angle of 125˚ (**Fig 3A**) to mimic the remodeled cephalic arch in patients after AFV placement. Inflow velocities ranging between 1–100 cm/s were tested. These include average flow rate measured in ESRD patients prior to AVF placement (5 cm/s) to peak flow rate measured in ESRD patients (70 cm/s) without stenosis [22]. Flow is simulated in COMSOL as both steady-state and pulsatile, using the 'Stationary' and 'Time Dependent'

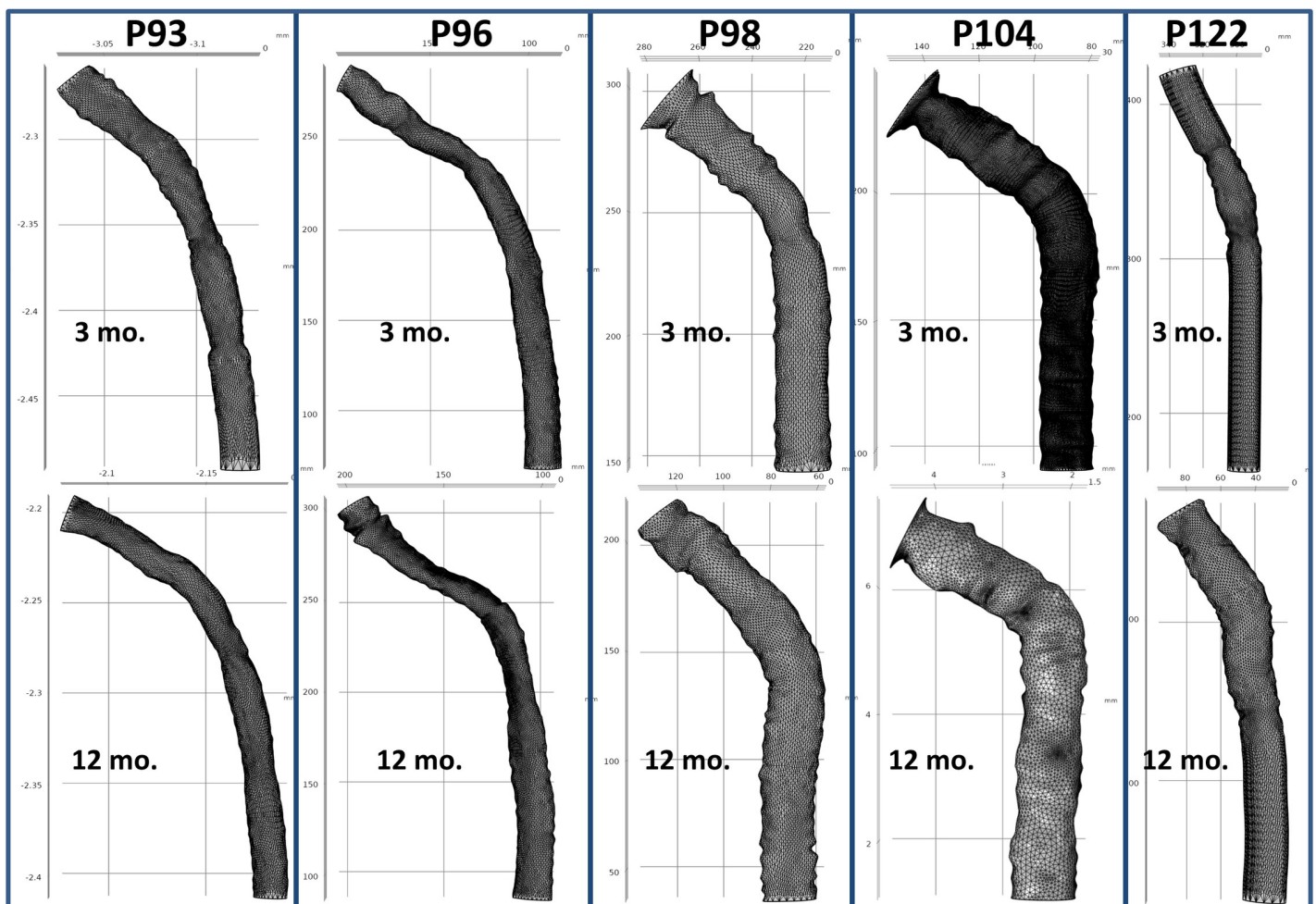

**Fig 3. Three dimensional models of cephalic arches in five ESRD patients obtained 3 months (top) and 12 months (bottom) after their AVF placement.** The models, displayed in the xy-plane, are reconstructed from venogram and IVUS imaging.

options, respectively. For steady-state flow, outlet pressure is set to 23 mm Hg, approximating the peak systolic pressure measured in patients' CA. Density, $\rho$ and viscosity, $\eta$ values used in the idealized model are approximated from whole blood: $\rho = 1{,}060\,\text{kg/m}^3$ and $\eta = 3.45$ mPa.s [20]. 'Fully developed flow' option is selected for both inlet and outlet, along with 'no-slip boundary' condition and 'incompressible flow' options. Though whole blood is shear thinning, it has been shown previously [23] that under similar conditions, it makes no appreciable difference whether blood is modeled as Newtonian or shear-thinning fluid. We use WBV measurements in our patient cohort for fixed $\eta$ in our simulations. The 'Physics' in COMSOL is described by the following equations: $\rho(\mathbf{u} \cdot \nabla)\mathbf{u} = \nabla \cdot \left[-p\mathbf{I} + \eta(\nabla\mathbf{u} + (\nabla\mathbf{u})^{T})\right] + \mathbf{F}$; and $\rho\nabla \cdot \mathbf{u} = 0$, where $\rho$ is density, $\eta$ is viscosity, $\mathbf{u}$ is the velocity vector, and $\mathbf{F}$ is the volume force vector, respectively. Physics-controlled meshes of coarse element sizes were generated using COMSOL. Mesh elements of variable size that match the local topology or surface roughness of 3D model of the CA were generated. We were able to obtain mesh convergence on a few, randomly selected CA models of patients and time-points. Parallel Sparse Direct and Multi-Recursive Iterative Linear Solver (PARDISO) in COMSOL with a relative tolerance of 0.001

was used for steady state flow. For pulsatile flow, an iterative solver, Generalized Minimum Residual (GMRES) was used with a residual tolerance of 0.01.

We simulated flow in the 3D models of all five patients' CAs reconstructed from IVUS and venogram images as described above, at two time-points each. Patient-specific peak systolic velocity and whole blood viscosity measured in our previous studies were used to simulate flow in all 10 models, along with the modules and parameters described above. Flow velocity and pressure profiles for each time point were calculated as output parameters for comparison. All other parameters were set as described for the idealized CA model. Shear stress was calculated as the product of shear stress function, 'spf.sr' in COMSOL and measured WBV for all cases. We also reported the cell Reynolds number parameter defined in COMSOL, $Re_{cell}$ using $\rho$ and $\eta$ described above, $u$ as the local flow velocity, and L as the size of the local mesh element. $Re_{cell,}$ can be conveniently used to compare flow within a CA model, across different patient models, time-points, flow conditions, etc.

To investigate geometric and flow characteristics within each model, we demarcated the CA of each subject into 'pre-bend', 'bend' and 'post-bend' regions and calculated the maximum velocity, pressure, and WSS from a randomly chosen 2D slice from each of these regions. Pre-bend was defined as the straight segment of the cephalic vein before the curve; bend was defined as the segment of the vein in between where it arches; post bend was defined as the straight segment of the cephalic vein before termination at the axillary vein (**Fig 1B**).

To simulate pulsatile flow, inlet flow velocity was applied as a sinusoidal wave to each CA reconstructed from each patient at 3 and 12 mo. time-points. For each model, the velocity amplitude was varied between peak systolic velocity, $u_{max}$ and diastolic velocity, $u_{min}$ at patient- and time-point specific pulse frequency, $f$ obtained from Doppler measurements, as $(u_{max}-u_{min})^*\sin(2\pi f)+u_{min}$. Outlet pressure was maintained at 20 mm Hg (3 mo.) or 23 mm Hg (12 mo.). An interval of 1.5 s was simulated in steps of 0.05 s for pulsatile flow; this time interval is in excess of the duration of a complete pulse cycle for all patients and time-points. Total time taken for each time-dependent simulation to complete was less than 1 hr. for all patients and time-points. All steady state simulations took < 10 min to run.

## Results

### Study population

Five patients who completed two protocol imaging procedures, blood flow velocity and viscosity measurements at 3 and 12 mo. were included in this investigation. Patient demographics and past history were obtained from electronic medical records (**Table 1**). A patient was considered to have coronary artery disease (CAD) if they had a myocardial infarction, or if cardiac catheterization showed significant coronary disease. A patient was considered to have

**Table 1. Patient characteristics.**

| Patient ID | Age (yr) | Sex | Ethnic | Weight (kg) | Height (cm) | BMI | Diabetic | HTN | CAD | PVD |
|---|---|---|---|---|---|---|---|---|---|---|
| 93 | 29 | M | AA | 142 | 167 | 50 | + | - | - | - |
| 96 | 34 | F | AA | 103.5 | 160 | 40.4 | + | + | - | - |
| 98 | 58 | M | AA | 99.3 | 175.3 | 32.3 | + | + | - | - |
| 104 | 23 | M | AA | 63.5 | 175.3 | 20.6 | - | - | - | - |
| 122 | 29 | M | AA | 113.4 | 182.9 | 33.9 | - | - | - | - |

yr = years; F = female; M = male; AA = African American; kg = kilogram; cm = centimeter; BMI = body mass index; HTN = hypertension; CAD = coronary artery disease; PVD = peripheral vascular disease; + = yes;— = no.

peripheral vascular disease (PVD) if they had a history of amputation from ischemic disease or if vascular study showed ankle-brachial index less than 0.9. We chose all subjects with the need for a primary BCF who were receiving hemodialysis three days a week. All patients were less than 60 years of age with no history of clinical vascular disease.

Peak systolic and diastolic velocities measured using Doppler, vitals, and calculated venous area, diameter, Reynolds number and arch angle are summarized in **Table 2**.

We noted significant changes in measured parameters local to the AVF, e.g., average vein diameter, and peak systolic and diastolic flow velocities in each patient at these two time-points, though overall blood pressure and pulse rates remained largely unchanged. There was local remodeling of the CA from the AVF maturation process, as shown by the increased mean cephalic vein diameters (0.85 ± 0.16 cm at 3 mo. to 0.87 ± 0.2 cm at 12 mo.). The mean arch angle of the CA, $\alpha$, as measured from venogram images decreased slightly (137 ± 11˚ at 3 mo. to 125 ± 17˚ at 12 mo.), indicating that the bend may grow more acute over time [24].

Blood flow velocities downstream to the AVF were measured in the subjects using Doppler imaging. We noted significant variability in the measured values across subjects, time-points and peak systolic and diastolic values, highlighting the dynamic nature of flow in the CA. The average flow velocity at 3 mo. (~45.2 ± 18.2 cm/s) was significantly higher (~50%) than the average at the 12 mo. time-point (~28.9 ± 20.1 cm/s).

Using measured vein diameter, flow velocity, and whole blood viscosity values (**Table 2**), we calculated *Re* for each subject and time-point (**Table 2**). As expected, there were large variations in *Re* between subjects and time-points, with an average of 1,239 ± 530 at 3 mo. and 740 ± 709 at 12 mo. We also noted overall difference among subjects, with patient P93 experiencing relatively low *Re* at both time-points ($Re_{avg}$ = 339) while we record much higher values for P122 ($Re_{avg}$ = 1,566).

## Simulating flow in 3D idealized computational models of cephalic arch

An idealized model of the CA was created to simulate flow under physiologic parameters associated with blood flow in patients with BCF access. Briefly, a cylindrical tube with a diameter of 0.95 cm (nominal vessel diameter in ESRD patients at QB0 = 0.3–0.8 cm; QB350 = 0.4–1.8 cm), and bent at 125˚ angle was used to model the CA in ESRD patients (**Fig 4A**). To recapitulate blood flow in the cephalic vein up the arm into the CA, we modeled flow entering through the end of the long arm of the CA model and exiting at the end of the short arm. For ease of

**Table 2. Patient measured and calculated parameters.**

| Patient ID: | Systolic velocity | Diastolic | WBV | BP | Pulse | Vein area | Vein diameter (cm) | Reynolds number | Arch angle, α |
|---|---|---|---|---|---|---|---|---|---|
| Time point | (cm/s) | velocity (cm/s) | (cP) | (mm Hg) | (beat/min) | (cm²) | | | (⁰) |
| P93: 3-month | 17.6 | 9.4 | 4.12 | 134/84 | 83 | 0.513 | 0.81 | 367 | 145 |
| P93: 12-month | 17.6 | 5.7 | 3.96 | 119/81 | 71 | 0.337 | 0.66 | 311 | 113 |
| P96: 3-month | 66.6 | 38.3 | 2.96 | 174/96 | 101 | 0.347 | 0.66 | 1574 | 133 |
| P96: 12-month | 43.5 | 21 | 4.08 | 122/70 | 81 | 0.337 | 0.66 | 746 | 132 |
| P98: 3-month | 49.7 | 38.2 | 3.2 | 130/76 | 90 | 0.532 | 0.82 | 1350 | 130 |
| P98: 12-month | 17.6 | 8.23 | 3.86 | 143/82 | 79 | 0.840 | 1.03 | 498 | 115 |
| P104: 3-month | 39.6 | 29.2 | 2.68 | 124/80 | 81 | 0.930 | 1.1 | 1723 | 125 |
| P104: 12-month | 9.4 | 4.6 | 4.63 | 125/81 | 84 | 0.594 | 0.9 | 194 | 115 |
| P122: 3-month | 52.5 | 24.3 | 3.96 | 99/60 | 88 | 0.559 | 0.84 | 1180 | 150 |
| P122: 12-month | 56.6 | 41.2 | 3.35 | 130/74 | 88 | 0.935 | 1.09 | 1952 | 152 |

WBV = whole blood viscosity; BP = blood pressure

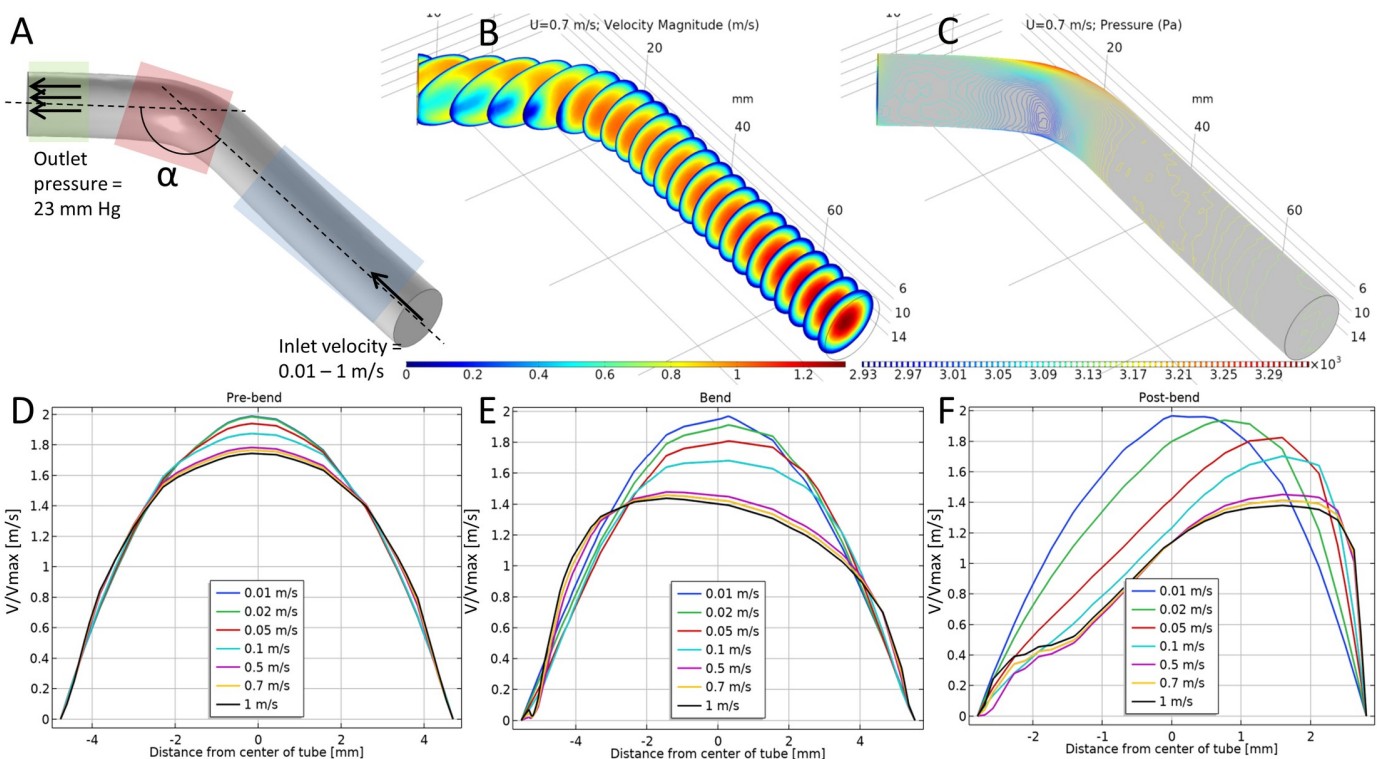

**Fig 4.** A) An idealized model of a CA in ESRD patient. The diameter of the model is 9.5 mm and bend angle, α is 125˚. The blue, red and green regions indicate the pre-bend, bend and post-bend regions, respectively. B) Simulated velocity and (C) pressure profiles, respectively in the 3D model under ESRD-specific physiological conditions. D-F) Velocity profiles across the CA model in the (D) pre-bend, (E) bend, and (F) post-bend regions in the idealized model across the tubular cross-section. Different inlet flow velocities range from healthy to ESRD values.

discussion we will call the longer arm of the model prior to the arch as 'pre-bend', the bent region as 'bend', and the shorter arm after the bend as 'post-bend' (**Fig 1, lower panel**). Initial conditions for the simulation were set as follows: inlet flow velocity, *u* was varied between 1–100 cm/s that included the flow velocities in the CA of healthy adults (5–10 cm/s) and patients with AVF (QB0 = 25–185 cm/s, QB350 = 25–288 cm/s). Flow velocity (**Fig 4B**) and pressure (**Fig 4C**) were plotted in 3D across the length of the idealized model at *u* = 70 cm/s. We saw lower flow rates and lower pressures along the inner wall of the bend and correspondingly higher velocities and higher pressure along the other wall, as indicated by the color of the velocity and pressure heatmaps. As endothelial activation is associated with areas of low WSS, we paid particular attention to this area as nidus for endothelial activation and clot formation.

Next, we plotted the normalized velocity profiles across the idealized model as function of distance from the center of the tubular cross-section with the tube's center set as zero (**Fig 4D–4F**). The velocities were normalized by the maximum velocity, $u_{max}$ in each case for ease of comparison. This was done for three arbitrary slices chosen from the pre-bend, bend and post-bend regions. For the pre-bend region (**Fig 4D**), the parabolic velocity profiles at low flow velocities (*u* < 10 cm/s) reflect Newtonian fluid under pipe-flow conditions [23, 25], but as *u* was increased to those seen in ESRD patients (*u* > 25 cm/s), the velocity profiles started to flatten and transition to plug-flow behavior of non-Newtonian fluids [23, 25]. The velocity profile changed from Newtonian (0.01 m/s) to shear-thinning (1 m/s) as flow velocity was increased and persisted into the 'bend' (**Fig 4E**) and 'post-bend' regions (**Fig 4F**). The velocity profiles also became non-axisymmetric in the 'bend' (with lower velocities along the outer wall, **Fig 4E**) and 'post-bend' (lower velocity along the inner wall, **Fig 4F**) regions at high velocities (*u* ≥ 0.5

m/s). This is expected for flow along bends, as the average distance traversed by the fluid along the inner wall of the bend is less than that along the outer wall; thus the velocity along the outer wall has to increase to compensate for the extra distance travelled for overall $u$ to remain constant (**Fig 4E**). Immediately into the post-bend region, the axial asymmetry switched direction (**Fig 4F**), such that the relative velocity along the inner wall was higher than that of the outer wall. These asymmetries in velocity profile lead us to predict two regions of abnormal WSS, one directly in the inner wall of the bend region, and the other at the beginning of the post-bend region, but along the outer wall of the CA. These predictions match a similar study in 2D [14].

## Simulating flow in 3D computational models of cephalic arch derived from patient data

Informed by general trends in flow characteristics obtained from simulations on the idealized models, we simulated flow in patient-specific models. Computational models of the CA at 3 and 12 mo. were used as the flow conduit. Although all subjects had primary outflow through the cephalic vein to support dialysis, we noted significant variations in the anatomic geometry between patients. In a previous investigation, we measured the inlet pressure in a cephalic arch (without obvious stenosis on a venogram) and found the average pressure to be 19.88 mm Hg at 3 mo. and 23.3 mm Hg at 12 mo. Using the patients' peak systolic velocity as the inlet velocity, WBV and average pressure measured in the CA as the outlet pressure at 3 and 12 mo., we plot the steady-state velocity (top) and pressure (bottom) profiles (**Fig 5**) in the 3D models of the CA for 5 patients at 3 and 12 mo. after the AVF placement. For all velocity and pressure profiles in **Fig 5**, the color indicates the magnitude and the color bar uses a rainbow scheme with blue denoting the lowest and red the highest values. The black tubes marked on the velocity profiles in **Fig 5** indicate the velocity field at that point, with the tube diameter being proportional to the shear rate. We noted considerable variation in the velocity and pressure profiles across the 5 subjects.

Non-Newtonian flow (asymmetric plug flow profile) was maintained along the bend and post-bend regions for all time-points. We observed low flow velocity and wall pressure along the inner wall of the bend in all cases (**Fig 5**), which may provide possible nidus of endothelial activation that lead to stenosis and thrombosis. For each patient and time point, we identified potentially problematic areas (e.g., constrictions and low WSS) referred to as Regions Of Interest (ROIs). We circled and numbered some of them in the 3D velocity profiles in **Fig 5**. We observed that ROIs that showed up at 3 mo. were likely to persist at the 12 mo. time-point, e.g., 1, 2 (**P93**), 4 (**P96**), 8 (**P104**), 9 (**P122**) in **Fig 5.** The ROIs that could be correlated with pre-exiting ROIs are marked as 1', 2', 4', 8', and 9', respectively in **Fig 5**. Two categories of ROI were seen in **Fig 5**: the first, associated with the narrowing of the vein diameter, as indicated by black dashed circles, e.g., 1, 5, 6, 9 and the second, areas of low flow velocities at the inner bend which may be associated with abnormal wall shear stress and highlighted by red dotted circles, e.g., 2, 4, 8, 10. The black streamlines denoting the velocity vector field overlaid on the velocity profiles show that all flow are largely laminar in the pre-bend and post-bend regions under steady-state flow. Non-laminar streamlines and recirculation zones appear in the bend regions under sufficiently high $Re$ ($Re >1,000$), e.g., P96, 3 mo., P104, 3 mo. and P122, 3 and 12 mo.

Much like $Re$ used to predict global flow characteristics, $Re_{cell}$ may be used to compare flow across the length of the CA, across different patients, flow conditions, time-points, etc. The $Re_{cell}$ values under steady state flow are shown in **S1 Fig, top panel** for all subjects at all time-points. WSS is plotted as a surface function for each model in **S1 Fig, bottom panel**. The values of $Re_{cell}$ and WSS are indicated as heat maps for each figure, where red indicates high and blue indicates low values, as before.

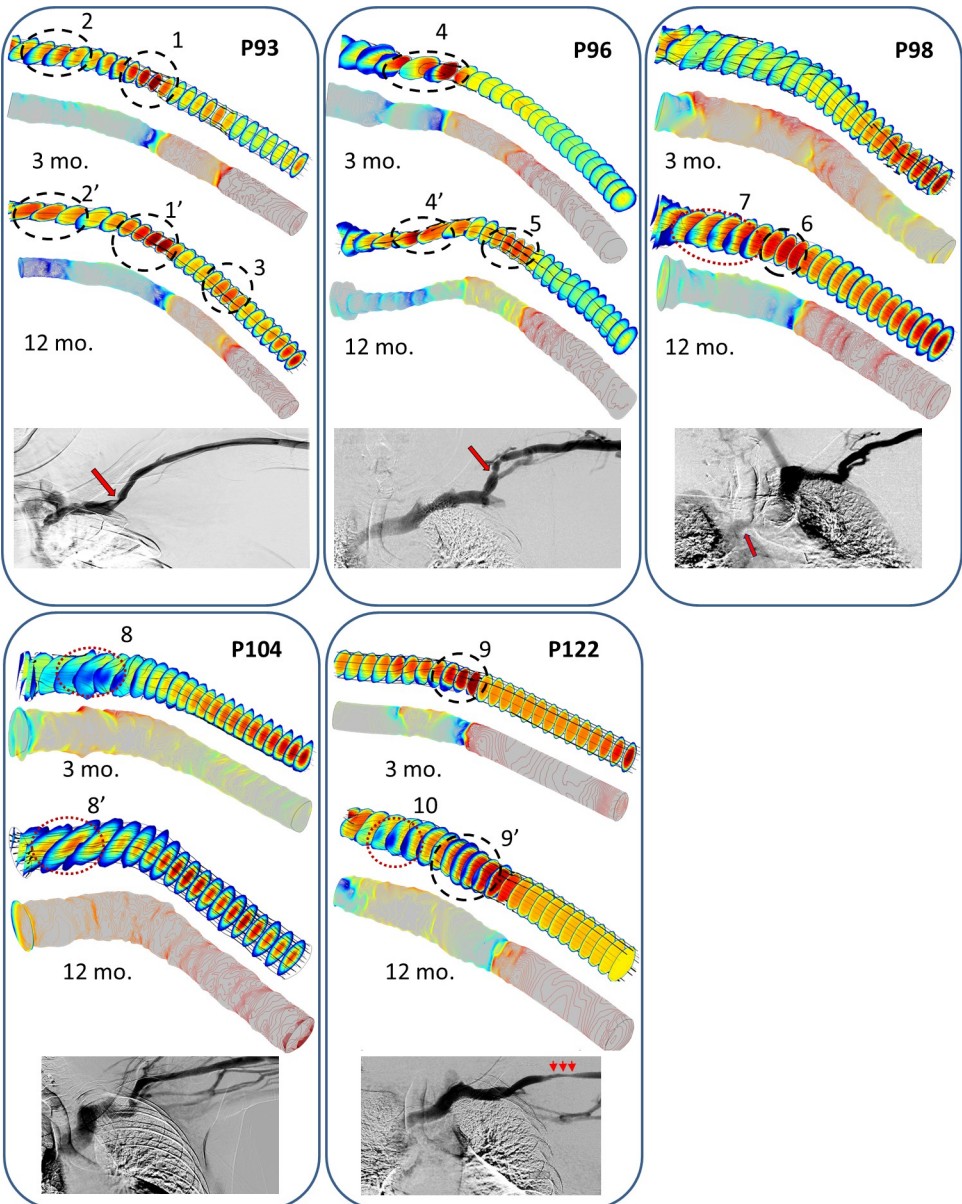

**Fig 5. Panels P93, P96, P98, P104 and P122 show the velocity and pressure profiles in 3D models constructed from IVUS and venogram measurements on the five ESRD patients' CA at 3 mo. (top), and 12 mo. (middle) and follow-up venogram (bottom) after AVF placement.** The color bar indicates the magnitude of velocity and pressure, respectively following a rainbow scheme, with blue denoting the lowest and red the highest magnitude. Black tubes superimposed on the velocity profiles indicate the velocity field at that point with the tube diameter being proportional to the shear rate. Flow parameters obtained from each patient's vitals and Doppler data were used to simulate flow.

In all cases, $Re_{cell}$ values were the highest at the center of the vein compared to the periphery, and increased at the bend of the CA, as expected in pipe flow. $Re_{cell}$ values also increased in regions where the vein narrowed, compared to regions upstream or downstream to it. Higher $Re_{cell}$ is associated with non-laminar flow and may lead to abnormal activation of endothelial cells lining the lumen. Interestingly, these regions also seemed to constrict further and develop stenosis over time (e.g., 12 mo. compared to 3 mo.). This may be due to the increasing activation of endothelial cells which acts as a feedback

mechanism to narrow the vein further, leading to higher velocities in the constriction and therefore, higher $Re_{cell}$.

To characterize flow in CA, we calculated the average diameter, velocity, pressure, and WSS from randomly chosen slices in the 'pre-bend', 'bend' and 'post-bend' regions. However, because of the high variability in patients' geometry and flow parameters, it was difficult to estimate meaningful averages of parameters such as velocity, diameter, etc. that can be generalized across patients and time-points. Instead, we analyzed each patient individually as function of time and correlated the results from the 3 and 12 mo. models with outcome, in the form of medical follow-up and venograms taken at later times. In some cases, the medical follow-ups extended across the entire duration of the five-year study (see ***Clinical follow up***).

## Pulsatile flow

Though blood flow in veins is not pulsatile under normal conditions, the local vein geometry and flow parameters after AVF placement get drastically remodeled over time [26, 27]. Not only do the vein diameter and blood flow velocity increase significantly, but the flow properties also change from primarily steady state to pulsatile flow [28], under the influence of the arterial flow created at the anastomosis. This is demonstrated by markedly different systolic and diastolic blood flow velocities recorded from Doppler measurements on mature AVF [29]. Interestingly, the effects of AVF maturation are not consistent among patients and strongly dependent on patient-specific physiological parameters. To simulate the effect of pulsatile flow in the idealized CA, we applied inlet flow velocity in the form of periodic waves (sinusoidal) at frequencies calculated from pulse rates measured from Doppler data.

Using pulse rates, blood pressure and peak systolic and diastolic blood flow velocities measured using Doppler, and assuming a sinusoidal flow pattern, we simulated time-dependent flow in the cephalic arch in patient P96 at 3 and 12 mo. using parameters described in **Methods.** The pulse frequency, $f$ measured using Doppler is reported in **Table 2**. The first panel plots the inlet velocity applied on the CA for patient P96 at 3 mo. (**Fig 6A**) and 12 mo. (**Fig 6B**) as a function of time, and sampled during the course of a pulse beat. Subsequent panels in **Fig 6A and 6B** show snapshots of the 3D velocity profile in the CA for patient P96 at 3 and 12 mo. (**Fig 6A** and **6B,** respectively) as the amplitude changes over the period of an oscillation. As before, the color bar shown in the last panel of each figure indicates the velocity magnitude in m/s and ranges from blue (lowest) to red (highest). The black tubes marked on the velocity profiles indicate the velocity field with the tube diameter indicating the shear rate at that point. In both **Fig 6A** and **6B**, we marked three time-points- 0.2, 0.4 and 0.8 s as '#', '+', and '*' in the first panel to show the inlet velocity applied to the CA and the corresponding 3D velocity profiles generated periodically in the CA for those inlet velocities during a pulse-beat. We noted that the areas of high velocity and low WSS seen at the bend at the 3 mo. time-point under steady state conditions (**Figs 5 and S1A, panels P96**) were recapitulated under pulsatile conditions at peak systolic values. As the inlet velocity, $u$ increased, this area of high velocity was seen to spread both upstream and downstream as function of $u$ and then dissipated, in a periodic fashion for every pulse, as $u$ was varied between peak systolic and diastolic. As the peak systolic velocity, $u = 0.67$ m/s at WBV = 2.96 mPa.s ($Re = 1,574$) was reached at ~ 0.4 s (marked as '+' in **Fig 6A**), we observed the periodic formation of a vortex at the bend of the CA, as shown by the black streamlines of the velocity field in a close-up of the bend in the last panel. The vortex dissipated as $u$ decreased to the diastolic value. Deviations from non-laminar flow, recirculation zones, vortices, etc. during pulsatile flow indicate that transient turbulent conditions may be generated in the CA during pulsatile flow that may lead to aberrant WSS and endothelial cell activation at high $u$.

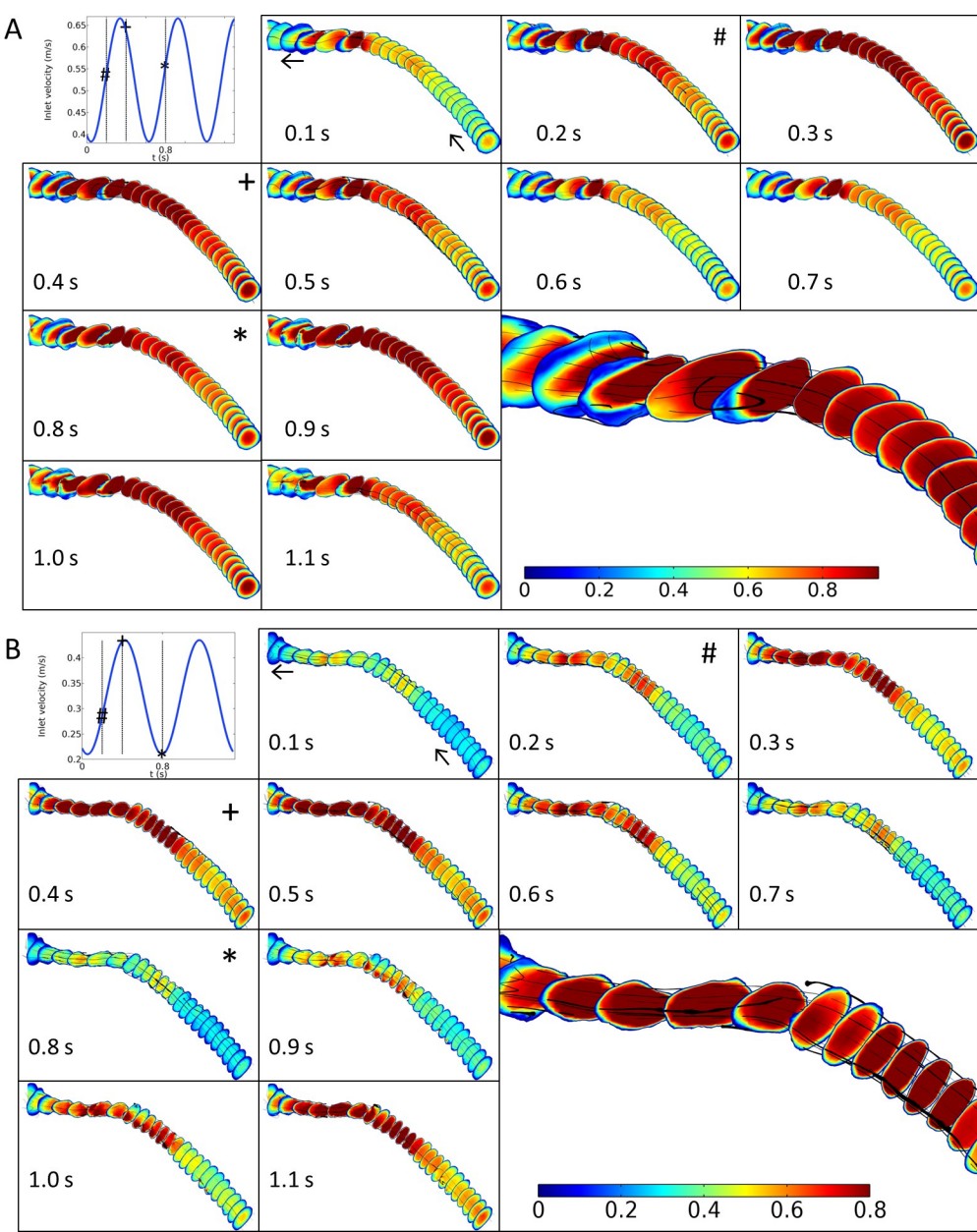

**Fig 6.** Each panel shows the velocity profile in the CA of patient P96 at (A) 3 mo. and (B) 12 mo. under pulsatile flow. In each figure, the first panel shows the inlet velocity as it cycles between peak systolic and diastolic as function of time sampled during pulsatile flow. The color bar in the last panel indicates the velocity magnitude. Black tubes marked on each velocity profile indicate the velocity field at that point with the tube diameter being proportional to the shear rate. Inlet velocities and their corresponding velocity profiles are highlighted at 0.2, 0.4 and 0.8 s, marked as '#', '+' and '*'. Arrows in the second panel indicate the direction of flow along the CA.

We saw similar trends in P96 under pulsatile flow for the 12-month time-point too: as the inlet velocity increased, the two areas at the constrictions (marked as 4' and 5 in **Fig 5, panel P96**) showed the most dramatic increase in velocity. The region between the constrictions also experienced marked increase in velocity as function of $u$ which then dissipated in a periodic fashion for every pulse, compared to the pre- and post-bend regions. Note that however, at comparatively modest $u = 0.44$ m/s and WBV = 4.08 mPa.s at 12 mo. (Re = 746), we do not see

any vortex formation in the velocity vector field at the bend in the patient's CA at peak $u$ (**Fig 6B,** last panel, close-up of the bend), unlike at 3 mo.

Similar 3D velocity profiles in the CA of patients P93 and P122 are shown as functions of time under pulsatile flow in **S2 Fig** (P93) and **S3 Fig** (P122) at 3 mo. (**A**) and 12 mo. (**B**), respectively. At overall low $u$ and high WBV, patient P93 showed largely laminar streamlines and moderate flow irregularities ($Re$ = 367, 311, respectively) at both 3 mo. (**S2A Fig**) and 12 mo. (**S2B Fig**) under sinusoidal flow. A constriction in the CA at the bend at 3 mo. developed slightly both upstream and downstream at 12 mo., causing moderate increase in flow velocities and shear rates in these regions. No vortex formations were noted in the velocity streamlines in the CA bend of P93 at the 3 mo. or 12 mo. time-points.

In contrast, patient P122, who had a constriction at the CA bend and also recorded high $u$ and high WBV at 3 mo. ($Re$ = 1,180; **S3A Fig**) and 12 mo. ($Re$ = 1,952; see last panel in **S3B Fig**) underwent considerable remodeling of the cephalic arch geometry between the 3 and 12 mo. time-points. P122 also showed high flow velocities with non-laminar streamlines in the bend region of the CA.

While the flow velocities in the CA from Doppler data can be well approximated as sinusoidal in most cases, we also modeled flow in the CA of patient P96 (3 mo.) as the periodic variation in inlet velocity as non-sinusoidal waveforms: square (**S4A and S4B Fig**), with duty cycle = 0.2 and 0.5, respectively) or saw-tooth (**S4C Fig**). The duty cycle indicates the fraction of one period in which the inlet velocity in maintained at peak systolic velocity. For example, at a duty cycle of 0.5, the flow velocities experience the peak systolic value for 50% of the time during a pulse beat. In contrast, at duty cycle = 0.2, the CA experiences peak systolic velocity for only 20% of the pulse duration. By changing the duty-cycle parameter, we were able to investigate the effect of longer dwell times at the peak systolic values may have on flow in the CA. The saw-tooth waveform in **S4C Fig** is interesting because the velocity waveform applied to the inlet has asymmetric leading and trailing edges: transition from diastolic to systolic velocities is slower than the transition from systolic to diastolic. When the rate of this transition from systolic to diastolic (or vice versa) is fast, we can see vortex formation even at comparatively low inlet velocity, e.g., at 0.5 s, marked as '×' in **S4C Fig**.

In all cases, the consequences of pulsatile flow were seen to broadly hold true across patients (e.g., P96, P122) where, at high $Re$, the velocity streamlines transition from laminar to non-laminar flow and back in the CA bend as $u$ varies between systolic and diastolic values. We posit that second-order effects, e.g., the time taken to transition between the peak systolic and diastolic values, may also influence non-linear flow behavior in the CA, leading to further abnormal endothelial cell activation and may be of interest in future studies.

## Simulations with decreased flow velocity

To further test the effect of high flow velocity and $Re$ on the velocity profiles and patient-specific ROI, we chose two instances in our dataset where abnormally high $u$ and $Re$ were recorded- P96 at 3 mo. (56.6 cm/s; 1,574) and P122 at 12 mo. (66.6 cm/s; 1,952) (Table 2). We simulated the flow profiles in the patient- and time-point-specific CA at 50%, 25% and 10% of the measured $u$. The streamlines in S5 Fig shows the velocity field indicated by black tubes in the patient's CA at 100% (original) and 50, 25 and 10% (reduced) $u$; the diameter of the tube is proportional to the shear rate at any point. ROIs 1 and 9' identified in Fig 5 previously are highlighted by dashed squares in S5A Fig, B for P96, 30 mo. and P122, 12 mo. respectively As expected, we see that the velocity field transitioned toward laminar flow as $u$ is reduced systematically. This is especially true in the highlighted ROIs when the velocity is reduced to 10% of the original value; the vortex in the ROI in P96, 3 mo. and turbulent streamlines in P122, 12

mo. are absent when u is reduced to 5.7 and 6.7 cm/s (Re = 157, 195) respectively. While other parameters such as vein diameter and blood viscosity also affect *Re*, we investigated *u* in particular because it varied the most in our patients (several folds) compared to the healthy baseline and is amenable to clinical intervention (see **Discussion**).

## Clinical follow up

The cohort exhibited outcomes ranging from patients having success with continued AVF use, to patients with recurrent thrombosis, cephalic arch stenosis, or steal syndrome. **Fig 5, bottom, in each panel** shows the follow up venograms taken between 21–60 months after AVF placement with correlation. The details of the heterogeneity in patients as the clinical follow up are provided. Our study highlights the variety of outcomes in each case (**Table 3**) that may be traced back to patient-specific modeling.

Recurrent venous stenosis and thrombosis of the AVF occurred in subject P93 requiring numerous interventional procedures to maintain fistula patency. At 48 months after AVF placement, a venogram was done for thrombosis and showed CAS was evident as shown by the red arrow in (**Fig 5, P93, bottom panel**) correlated with areas of high velocity both at 3 months (**2**) and 12 months (**2$^{'}$**) (**Fig 5, P93, top and middle panels**). P93 eventually developed severe central venous stenosis in the brachiocephalic vein and juxta-arterial anastomotic stenosis, requiring angioplasty and stent placement (not shown). The patient eventually lost patency of the BCF and needed catheter access and a new AVF in the opposite arm.

At 21 months, high venous pressures were recorded in P96 during hemodialysis with pseudo-aneurysms at the cannulation areas. Venogram imaging showed that the patient had developed venous outflow stenosis and CAS (not shown). At 36 months there was evidence of significant CAS at the bend of the arch (**Fig 5**, P96 bottom panel) which corresponded to the abnormal high flow profiles (**4** and **4$^{'}$**). (**Fig 5, P96, top and middle panels**).

P98 required a follow up venogram at 60 months for swelling of the arm. The venogram showed stenosis at the junction of the brachiocephalic vein and superior vena cava, indicated by the red arrow (**Fig 5, P98, bottom panel**) with retrograde flow up the internal jugular vein and small axillary collateral. The flow and pressure profiles at three months do not show high flow and pressure. The 12 month velocity profile is high (**6** and **7**) which correlates anatomically with collaterals evident on the follow up venogram.

P104 developed dual flow through the cephalic and basilic vein systems with most flow going through the basilic system. A vascular lab study at month 28 showed a volumetric flow of 3,336 ml/min in the basilic vein and a volumetric flow of 686 ml/min in the cephalic vein. This process of flow diversion (cephalic vein to basilic vein) may have been evident as early as 12 months after AVF, as shown by the low peak systolic velocity (**Table 2**). The patient subsequently developed a very large aneurysm of the basilic vein which required a basilic vein

**Table 3. Clinical outcomes of patients.**

| Patient ID | CAS | AVF Loss | Other |
|---|---|---|---|
| 93 | + | + | Recurrent thrombosis, stenosis of anastomosis, venous outflow and central veins |
| 96 | + | - | Recurrent stenosis, aneurysms |
| 98 | - | - | Central venous stenosis |
| 104 | - | - | Dual flow basilic and cephalic veins with aneurysms required surgical resection |
| 122 | + | + | Venous outflow stenosis and steal syndrome required AVF ligation |

CAS = cephalic arch stenosis; AVF = arterioveous fistula; + = yes;— = no.

ligation and aneurysm resection. The high flow velocities at 3 and 12 months (**Fig 5**, **P104, 8** and **8'**) correlate with a collateral vein imaged at the later time point.

CAS developed in P122, as shown by the red arrows, was first observed at 24 months (**Fig 5**, **P122, bottom panel**) which correlated in location to high flow velocities at 3 months (**9**) and 12 months (**9'**). P122 also developed significant venous outflow stenosis which required angioplasty and later went on to develop a severe steal syndrome which resulted in AVF abandonment. The clinical outcomes and follow up above are also summarized in Table 2.

## Discussion

In the current study, we successfully modeled the CA in 3D using two independent and orthogonal imaging techniques, viz., IVUS and venogram. Being able to visualize the model in 3D helped identify stenosis and other topological irregularities that may not be directly evident in a 2D image taken on a single, fixed plane. We observed the overall geometry for the CA, including average vein diameter, arch angle, and irregularities in the vein walls to be highly variable between subjects that can be associated with pathologic stenosis at later time points, e.g., P93, P96 and P122. This heterogeneity coupled with variations of hemodynamic parameters of each patient, e.g., velocity, viscosity, pulsatility, etc., measured independently through Doppler and viscosity tests, make the CA an interesting but complex system to study. Individualized computational modeling can be of great importance in understanding the contribution of each of the above parameters to the CA remodeling and/or AVF failure. While computational modeling has been constructed from IVUS images in coronary lesions [30] and used to determine WSS at the anastomotic angle in AVF [7], it has not been used to look at the effect of the AVF on flow downstream. Because of the unique geometry and dynamic remodeling of the CA as a direct consequence of the AVF placement, the CA can be an area of pathological and interventional relevance in context of ESRD and hemodialysis. This is the first reported computational fluid dynamics simulation on a series of human patients with AVF access studied over time.

Using 3D computational modeling, we showed that abnormal hemodynamics, specifically high flow velocity in the CA, predicted areas that develop into stenosis or have abnormal collateral vessel formation. To test the effect of fluid flow in a simplified CA and decouple the effects of local vein topology/surface roughness and stenosis from vein diameter and flow velocities, we simulated blood flow in a 3D idealized vein model with dimensions similar to the CA in ESRD patients under hemodialysis. We observed that when the velocity increased from healthy values to the numbers measured in ESRD patients, the velocity profile across the vein cross-section changed from Newtonian to non-Newtonian. At the bend, the asymmetry in the velocity profile across the vein cross-section, with flow velocities higher along the outer wall and lower close to the inner wall. In the post-bend region, the idealized model again showed asymmetry in velocity profiles but with flow velocities higher along the inner wall and lower along the outer wall. These findings identified some general trends to investigate in patient-specific 3D models of the CA. The idealized CA model shown in **Fig 4** with its smooth surface and uniform diameter lacks the intricate topographic changes found in the individualized models that greatly affect the hemodynamics.

Conventional angiography remains the main imaging modality used for vascular imaging. Traditionally, AVF complications have been diagnosed and treated with endovascular techniques most commonly by percutaneous angiography techniques that yield 2D images of the vessel of interest. As shown in our study, IVUS adds valuable information about the actual vessel's cross-sectional area showing areas of hemodynamic turbulence and venous stenosis that may not be otherwise appreciated from 2D venograms.

3D modeling of the CA was performed on five subjects at two time points: one at 3 mo. that represent a mature AVF, and another at 12 mo. at which restoration of WSS should occur [31]. The remodeling response in the vein after AVF creation should be greatest in the first 2–6 weeks, during which maturation is achieved. Beyond this time, the vein is found to be less "shear responsive" and adapted to the new flow environment, given venous stenosis does not occur [32]. Our 3D models suggest that the CA is an area of active remodeling beyond 3 mo., there being significant differences between the 3 and 12 mo. models for all 5 patients in our study. In fact, follow-up venograms of the CA (as late as 60 months from AVF placement) suggest that the CA continues to remodel over time.

Using baseline blood flow velocities measured in the patient cohort using Doppler, we identified areas of abnormal blood flow in our models at 3 and 12 mo. such as high velocity and low WSS. Correlating these areas between the two time-points, follow-up venograms taken at later times for the patients and outcome, we find that some areas of abnormal flow that show up as early as 3 mo. are seen to persist at 12 mo. and can be associated with stenosis. In addition, the areas of low WSS and high pressure at 3 and 12 mos. are predictive of venous stenosis. We show the venous bends of the arch have anisotropic (low) velocity along the inner bend, low WSS and high pressure. In a blood vessel, velocity of blood flow increases after a constriction or venous stenosis. The velocity and diameter in our study may not show a relationship because of both upstream (central vein) and downstream (anastomotic) stenosis.

Patients are subject to further increase in blood flow during hemodialysis three times a week that is estimated to be 18% above the AVF baseline blood flow [33]. Elevated velocities during dialysis lead to increased $Re$ and create turbulent conditions in the CA, affecting the endothelium and worsening the stenosis. High blood flow is prescribed during hemodialysis with shortened time to maximize solute clearance may put the AVF at risk for adverse outcomes [34]. The optimal blood flow during hemodialysis that can maintain adequate clearance without contributing to AVF failure is not known and warrants further investigation.

There are some limitations to our study which include a descriptive study in a small cohort of patients lacking racial diversity; the subjects were mostly males who were doing well without additional co-morbid conditions. None of the subjects required hospitalization during the study, and attended most hemodialysis sessions. While the sample size is small, but we do descriptive, numerical and quantitative analysis with the overall aim to demonstrate the feasibility of computational modeling show abnormal flow dynamics in the CA. Another limitation is that the current IVUS data were obtained only from the CA, as this was the area of interest at the time under investigation; we do not have images of the anastomosis and central veins for this cohort, which could further influence flow velocity and resultant hemodynamics. There are subtle changes in the CA such as constrictions which alter the hemodynamics and endothelial activation. There are also additional vessels known as collaterals (evident in venograms) that can divert flow away from the CA which affect the hemodynamics but are not accounted for in our model. Finally, and perhaps most importantly, our models did not account for the effect of biochemical properties on the system, such as interplay between the blood components, endothelium, the coagulation cascade, and inflammatory response from the abnormal flow in the CA. Biochemical factors and cellular remodeling are also important in understanding and preventing complications and AVF failure and thus worthy of further investigation.

In summary, we have shown a systematic approach to exploring complications in individual ESRD patients under hemodialysis that was never been attempted before to construct a computational model of the cephalic arch using IVUS and 2D venograms. We emphasize the importance of the CA geometry and abnormal hemodynamics, particularly in the bend, that make the CA downstream to an AVF a unique area for study. Our modeling shows that the signals for failure may come along early (3 mo.) and if left untreated, can become serious.

Through systematic alterations of geometric or flow parameters, this modeling technique may also be used to prevent flow conditions that deviate from steady laminar flow which contributes to endothelial dysfunction. To maintain laminar flow and endothelial health, low *Re* may be necessary and achieved in lower flow states by altering the anastomotic angle when the AVF is created [35, 36]; banding the arterial inflow [37]; creation of an anastomosis by endovascular technique or the SLOT technique which both offer a lower flow state [38, 39]; or limiting the length of the arteriotomy when the BCF is created [40]. These computational models lay the foundation for a study to determine and optimize the flow and pressure parameters which could prevent altered hemodynamics that cause AVF complications and failure. Using computational data derived from the protocols outlined in this paper, velocity and pressure could be optimized at the anastomosis by external devices such as Optiflow [35] and VasQ [36] or internal devices such as stents [41] which are designed to minimize flow disturbances with consequences on the CA. Computational modeling shows that hemodynamics may be able to predict clinical sequela, but this will not provide the whole picture. The endothelium and blood components are also important which need to be integrated into a model of the cephalic arch. Future studies will include prospective imaging with computational modeling on a greater number of subjects with statistical tests of the prognostic performance.

## Supporting information

**S1 Video. Velocity profile in the cephalic arch P93 at 3 months.**
(AVI)

**S2 Video. Velocity profile in the cephalic arch P93 at 12 months.**
(AVI)

**S3 Video. Velocity profile in the cephalic arch P96 at 3 months.**
(AVI)

**S4 Video. Velocity profile in the cephalic arch P96 at 3 months.**
(AVI)

**S5 Video. Velocity profile in the cephalic arch P98 at 3 months.**
(AVI)

**S6 Video. Velocity profile in the cephalic arch P98 at 3 months.**
(AVI)

**S7 Video. Velocity profile in the cephalic arch P104 at 3 months.**
(AVI)

**S8 Video. Velocity profile in the cephalic arch P104 at 3 months.**
(AVI)

**S9 Video. Velocity profile in the cephalic arch P122 at 3 months.**
(AVI)

**S10 Video. Velocity profile in the cephalic arch P122 at 3 months.**
(AVI)

**S1 Fig. Each panel shows the $Re_{cell}$ (top) and WSS (bottom) values in the 3D models of the five ESRD patients' CA at 3 and 12 mo. after AVF placement.** The color indicates the magnitude of $Re_{cell}$ and WSS, respectively, following a rainbow scheme, with blue denoting the lowest and red the highest values.
(TIF)

**S2 Fig.** The velocity profile in the CA of patient P93 at (A) 3 months and (B) 12 months under pulsatile flow are shown at different times as indicated. The first panel in each figure shows the inlet velocity as it increases (systolic) and decreases (diastolic) as function of time. Inlet velocities and their corresponding velocity profiles are marked as '#', '+' and '*' at three time-points, 0.2, 0.4 and 0.8 s. The color bar in the last panel indicates the lowest velocity in blue and the highest in red. Arrows in the second panel indicate the direction of flow along the CA.
(TIF)

**S3 Fig.** The velocity profiles in the CA of patient P122 at (A) 3 mo., and (B) 12 mo. under pulsatile flow are shown at different time-points. The first panel shows the inlet velocity as it increases (systolic) and decreases (diastolic) as function of time. Inlet velocities and their corresponding velocity profiles are highlighted at three time-points, 0.2, 0.4 and 0.8 s, marked as '#', '+' and '*'. The color bar in the last panel indicates the lowest velocity in blue and the highest in red. Arrows in the second panel indicate the direction of flow along the CA.
(TIF)

**S4 Fig. The velocity profile in the CA of patient P96 at 3 months under pulsatile flow.** Each subfigure shows the inlet velocity applied as (**A**) a square wave with duty cycle = 0.5, (**B**) square wave with duty cycle = 0.2, and (**C**) a saw-tooth wave. As before, the first panel shows the inlet velocity as it increases (systolic) and decreases (diastolic) as function of time. Inlet velocities and their corresponding velocity profiles are highlighted at three time-points, 0.2, 0.4 and 0.8 s, marked as '#', '+' and '*' in **A** & **B**, and four time-points, 0.2, 0.4, 0.5, 0.6 and 0.8 s, marked as '#', '+', '×', '♣' and '*' in **C**. The color bar in the last panel indicates the lowest velocity in blue and the highest in red. Arrows in the second panel indicate the direction of flow along the CA.
(TIF)

**S5 Fig. Velocity fields in patients' CA show the transition to laminar flow with decreased flow velocity.** Black tubes indicate the velocity field at that point, with the tube diameter being proportional to the shear rate. Inlet velocities used in simulating flow in each panel are at 100%, 50%, 25% and 10% of the measured systolic velocity, $u$ in Table 2. All other flow parameters are obtained from each patient's vitals and Doppler data. The ROI for the patient and time-point are highlighted in dashed squares. (**A**) Patient P96 at 3 mo.; (**B**) Patient P122 at 12 mo.
(TIF)

**S1 File.**
(PDF)

## Acknowledgments

We thank Travis Bee for helpful advice on 3D modeling. We thank all the patients who gave their time for the procedures.

## Author Contributions

**Conceptualization:** Mary Hammes, Andres Moya-Rodriguez, Cameron Bernstein, Sandeep Nathan, Anindita Basu.

**Data curation:** Mary Hammes, Andres Moya-Rodriguez, Cameron Bernstein, Sandeep Nathan, Rakesh Navuluri, Anindita Basu.

**Formal analysis:** Mary Hammes, Andres Moya-Rodriguez, Rakesh Navuluri, Anindita Basu.

**Funding acquisition:** Mary Hammes.

**Investigation:** Mary Hammes, Cameron Bernstein, Sandeep Nathan, Anindita Basu.

**Methodology:** Mary Hammes, Andres Moya-Rodriguez, Cameron Bernstein, Sandeep Nathan, Rakesh Navuluri, Anindita Basu.

**Project administration:** Mary Hammes.

**Resources:** Mary Hammes, Anindita Basu.

**Software:** Andres Moya-Rodriguez, Cameron Bernstein, Anindita Basu.

**Supervision:** Mary Hammes, Anindita Basu.

**Validation:** Mary Hammes, Andres Moya-Rodriguez, Cameron Bernstein, Sandeep Nathan, Anindita Basu.

**Visualization:** Mary Hammes, Rakesh Navuluri, Anindita Basu.

**Writing – original draft:** Mary Hammes, Anindita Basu.

**Writing – review & editing:** Mary Hammes, Andres Moya-Rodriguez, Cameron Bernstein, Sandeep Nathan, Rakesh Navuluri, Anindita Basu.

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
