## [Decision Letter · Decision Letter 0]

18 May 2021

PONE-D-21-10937

A novel computational model of the cephalic arch predicts hemodynamic profiles in patients with brachiocephalic fistula access receiving hemodialysis

PLOS ONE

Dear Dr. Hammes,

Thank you for submitting your manuscript to PLOS ONE. After careful consideration, we feel that it has merit but does not fully meet PLOS ONE’s publication criteria as it currently stands. Therefore, we invite you to submit a revised version of the manuscript that addresses the points raised during the review process.

We look forward to receiving your revised manuscript.

Kind regards,

Fang-Bao Tian

Academic Editor

PLOS ONE

Journal Requirements:

2.Your ethics statement should only appear in the Methods section of your manuscript. If your ethics statement is written in any section besides the Methods, please delete it from any other section.

3.Thank you for stating the following in the Acknowledgments Section of your manuscript:

"A. M-R was supported by the NSF GRFP

fellowship. The work was partly funded by the Ginny and Simon Aronson Research Award,

University of Chicago Institute of Translational Medicine Pilot Award, and A.B.’s research

development funds."

"This publication was made possible by the National Institute of Diabetes and Digestive Diseases (NIDDK) and the National Institutes of Health (NIH) under award number RO1DK090769. "

Additional Editor Comments:

Thank you for resubmitting this work. I have invited a third person to give comments on the revision, who has a few major concerns. I would encourage you to carefully consider these comments, and revise the manuscript for further consideration.

Reviewers' comments:

Reviewer's Responses to Questions

**Comments to the Author**

1. Is the manuscript technically sound, and do the data support the conclusions?

Reviewer #1: Partly

2. Has the statistical analysis been performed appropriately and rigorously? 

Reviewer #1: Yes

3. Have the authors made all data underlying the findings in their manuscript fully available?

Reviewer #1: Yes

4. Is the manuscript presented in an intelligible fashion and written in standard English?

Reviewer #1: Yes

5. Review Comments to the Author

Reviewer #1: The overall aim of this work is to demonstrate the feasibility of computational modelling to unveil abnormal flow dynamics in the CA that are unique to individual patients. Five patient-specific models are reconstructed and considered by using commercial software. However, it is not convincing for me to accept it as the innovation as similar research has been conducted before. Even machine learning method has been considered to establish a model for the prediction of a similar hemodynamic problem. The major conclusion from this work is claimed to prove the importance of patient-specific model and the feasibility of using numerical simulation to resolve the flow dynamics, which is not brand new and sounds lacking deeper information. Actually, this point also agrees with the comments of Review 2. The patient-specific numerical simulation has been conducted, but detailed analysis is missing. I suggest the author to do more analysis in terms the change of the stenosis, and possible to provide some insights for the therapy.

6. PLOS authors have the option to publish the peer review history of their article (what does this mean?). If published, this will include your full peer review and any attached files.

Reviewer #1: No

---

## [Author Response · Author response to Decision Letter 0]

1 Jun 2021

PONE-D-21-10937

A novel computational model of the cephalic arch predicts hemodynamic profiles in patients with brachiocephalic fistula access receiving hemodialysis

PLOS ONE

Journal Requirements:

Response: We reviewed the PLOS ONE’s style requirements and formatted our manuscript accordingly. 

2.Your ethics statement should only appear in the Methods section of your manuscript. If your ethics statement is written in any section besides the Methods, please delete it from any other section. 

Response: The ethics statement is written in the first paragraph of the Methods section. 

3.Thank you for stating the following in the Acknowledgments Section of your manuscript: 

"A. M-R was supported by the NSF GRFP fellowship. The work was partly funded by the Ginny and Simon Aronson Research Award, University of Chicago Institute of Translational Medicine Pilot Award, and A.B.’s research development funds." We note that you have provided funding information that is not currently declared in your Funding Statement. However, funding information should not appear in the Acknowledgments section or other areas of your manuscript. We will only publish funding information present in the Funding Statement section of the online submission form.

"This publication was made possible by the National Institute of Diabetes and Digestive Diseases (NIDDK) and the National Institutes of Health (NIH) under award number RO1DK090769. "

Response: We have deleted the statement, “A. M-R was supported by the NSF GRFP fellowship. The work was partly funded by the Ginny and Simon Aronson Research Award, University of Chicago Institute of Translational Medicine Pilot Award, and A.B.’s research development funds” from the Acknowledgements and would like this statement to be included in the Funding Statement. 

Additional Editor Comments:

Thank you for resubmitting this work. I have invited a third person to give comments on the revision, who has a few major concerns. I would encourage you to carefully consider these comments, and revise the manuscript for further consideration.

Reviewers' comments:

Reviewer's Responses to Questions

Comments to the Author

1. Is the manuscript technically sound, and do the data support the conclusions?

Reviewer #1: Partly

Response: We have added in the discussion the limitations of the small sample size We provide in-depth descriptive, numerical and quantitative analysis. The future work will be to prospectively look at a large cohort of subjects to evaluate the statistical significance of computational modeling to predict clinical outcomes. 

2. Has the statistical analysis been performed appropriately and rigorously?

Reviewer #1: Yes

3. Have the authors made all data underlying the findings in their manuscript fully available?

Reviewer #1: Yes

4. Is the manuscript presented in an intelligible fashion and written in standard English?

Reviewer #1: Yes

5. Review Comments to the Author

Reviewer #1: The overall aim of this work is to demonstrate the feasibility of computational modelling to unveil abnormal flow dynamics in the CA that are unique to individual patients. Five patient-specific models are reconstructed and considered by using commercial software. However, it is not convincing for me to accept it as the innovation as similar research has been conducted before. Even machine learning method has been considered to establish a model for the prediction of a similar hemodynamic problem. The major conclusion from this work is claimed to prove the importance of patient-specific model and the feasibility of using numerical simulation to resolve the flow dynamics, which is not brand new and sounds lacking deeper information. Actually, this point also agrees with the comments of Review 2. The patient-specific numerical simulation has been conducted, but detailed analysis is missing. I suggest the author to do more analysis in terms the change of the stenosis, and possible to provide some insights for the therapy. 

Response: We thank the reviewer for this comment. We agree with these comments and have removed the word ‘novel’ in the title as others have reconstructed hemodynamics using computational software. 

Previous computational work has been done as discussed in the Introduction with references 9-11, although our work moves the field forward as we use actual images obtained from IVUS and venogram in patient-specific models in 3D with along with computational modeling of flow using parameters from Doppler imaging, blood viscosity measurements, and patients’ vitals recorded at the time of different imaging used in the study. The 5 subjects studied had models created at 3 and 12 months with obvious hemodynamic changes depicted most significantly on the lower border of the bend in the cephalic arch. Follow-up clinical venograms at later time points (up to 5 years) showed significant stenosis with anatomic correlation. We have shown that changes as early as 3 months may be connected to later venous stenosis. 

The comments provided by the reviewer above have prompted us to discuss potential therapeutic solutions to the future complications, based on our modelling. The hemodynamic changes and resultant wall shear stresses are largely influenced by high flow velocity in the cephalic arch. Our hypothesis is that a reduction in flow velocity will change the turbulent flow toward laminar flow. We tested this by reducing the flow to 50%, 25% and 10% of the original velocities measured in two extreme cases (P96 at 3 months and P122 at 12 months) and have added these results to a new sub-section, ‘Simulations with decreased flow velocity’ along with a new supplementary figure (Figure S5). This can provide direct insights for therapy as reduced flow (more laminar) may prevent the complications caused by hemodynamic turbulence. Lower flow velocities could be achieved by limiting flow when the BCF is created by 1) limiting the size of the arteriotomy (reference 40 was added); 2) changing the angle of the anastomosis with devices such as a VasQ (reference 36); or 3) the design of the type of anastomosis to promote lower flow (references 38, 39 were added). These insights for the future were included in the discussion. 

6. PLOS authors have the option to publish the peer review history of their article (what does this mean?). If published, this will include your full peer review and any attached files.

Do you want your identity to be public for this peer review? For information about this choice, including consent withdrawal, please see our Privacy Policy.

Reviewer #1: No

---

## [Decision Letter · Decision Letter 1]

18 Jun 2021

Computational modeling of the cephalic arch predicts hemodynamic profiles in patients with brachiocephalic fistula access receiving hemodialysis

PONE-D-21-10937R1

Dear Dr. Hammes,

We’re pleased to inform you that your manuscript has been judged scientifically suitable for publication and will be formally accepted for publication once it meets all outstanding technical requirements.

Kind regards,

Fang-Bao Tian

Academic Editor

PLOS ONE

Additional Editor Comments (optional):

Reviewers' comments:

Reviewer's Responses to Questions

**Comments to the Author**

1. If the authors have adequately addressed your comments raised in a previous round of review and you feel that this manuscript is now acceptable for publication, you may indicate that here to bypass the “Comments to the Author” section, enter your conflict of interest statement in the “Confidential to Editor” section, and submit your "Accept" recommendation.

Reviewer #1: All comments have been addressed

2. Is the manuscript technically sound, and do the data support the conclusions?

Reviewer #1: Yes

3. Has the statistical analysis been performed appropriately and rigorously? 

Reviewer #1: Yes

4. Have the authors made all data underlying the findings in their manuscript fully available?

Reviewer #1: No

5. Is the manuscript presented in an intelligible fashion and written in standard English?

Reviewer #1: Yes

6. Review Comments to the Author

Reviewer #1: A conclusion of this work should be given at the end of this paper.

The overall aim of this work is to demonstrate the feasibility of computational modelling to unveil abnormal flow dynamics in the CA that are unique to individual patients. The potential therapeutic solutions to the future complications based on the modelling is also discussed.

7. PLOS authors have the option to publish the peer review history of their article (what does this mean?). If published, this will include your full peer review and any attached files.

Reviewer #1: No

---

## [Editor Report · Acceptance letter]

5 Jul 2021

PONE-D-21-10937R1 

Computational modeling of the cephalic arch predicts hemodynamic profiles in patients with brachiocephalic fistula access receiving hemodialysis 

Dear Dr. Hammes:

I'm pleased to inform you that your manuscript has been deemed suitable for publication in PLOS ONE. Congratulations! Your manuscript is now with our production department. 

Kind regards, 

on behalf of

Dr. Fang-Bao Tian 

Academic Editor

PLOS ONE